# Cytoskeletal Responses and Aif-1 Expression in Caco-2 Monolayers Exposed to Phorbol-12-Myristate-13-Acetate and Carnosine

**DOI:** 10.3390/biology12010036

**Published:** 2022-12-25

**Authors:** Aurora Mazzei, Patrizia Pagliara, Gianmarco Del Vecchio, Lucia Giampetruzzi, Francesca Croce, Roberta Schiavone, Tiziano Verri, Amilcare Barca

**Affiliations:** 1Department of Biological and Environmental Sciences and Technologies (DeBEST), University of Salento, 73100 Lecce, Italy; 2Institute for Microelectronics and Microsystems IMM-CNR, Via per Monteroni “Campus Ecotekne”, 73100 Lecce, Italy

**Keywords:** intestinal epithelial monolayers, Caco-2 cells, actin cytoskeleton, allograft inflammatory factor 1, carnosine

## Abstract

**Simple Summary:**

The functionality of the enterocyte monolayer is directly impaired by inflammatory insults targeting the major cellular processes, including the cytoskeletal dynamics involving actin elements. Enterocyte’s actin cytoskeleton plays key roles in maintaining the epithelial monolayer’s integrity, and its remodeling is critically intertwined with the transition from physiological to pathological states of the gastrointestinal epithelial barrier challenged by inflammation onsets. Hence, understanding the behavior of the actin cytoskeleton in enterocytes forming the epithelial monolayer is a primary aim for clarifying fundamental aspects of inflammatory mechanisms in the gastrointestinal tract. Here, we analyzed the changing aspects of cytoskeletal actin in the human Caco-2 epithelial cell model at two different stages of differentiation: undifferentiated cells and spontaneously differentiated enterocyte-like cells. An in vitro inflammation-mimicking stimulus (phorbol-12-myristate-13-acetate) was used for challenging intestinal epithelial cells in association with the naturally occurring carnosine dipeptide, which showed its potential counteraction against alterations of the actin cytoskeleton in the enterocyte-like monolayers. Through such experiments, for the first time, we described in enterocyte-like monolayers the expression, localization, and variations of the allograft inflammatory factor 1, a protein functionally related to both inflammatory and cytoskeletal pathways, that we suggest considering as interesting features potentially marking the intestinal epithelial monolayers.

**Abstract:**

The dis(re)organization of the cytoskeletal actin in enterocytes mediates epithelial barrier dys(re)function, playing a key role in modulating epithelial monolayer’s integrity and remodeling under transition from physiological to pathological states. Here, by fluorescence-based morphological and morphometric analyses, we detected differential responses of cytoskeletal actin in intestinal epithelial Caco-2 cell monolayers at two different stages of their spontaneous differentiation, i.e., undifferentiated cells at 7 days post-seeding (dps) and differentiated enterocyte-like cells at 21 dps, upon challenge in vitro with the inflammation-mimicking stimulus of phorbol-12-myristate-13-acetate (PMA). In addition, specific responses were found in the presence of the natural dipeptide carnosine detecting its potential counteraction against PMA-induced cytoskeletal alterations and remodeling in differentiated Caco-2 monolayers. In such an experimental context, by both immunocytochemistry and Western blot assays in Caco-2 monolayers, we identified the expression of the allograft inflammatory factor 1 (AIF-1) as protein functionally related to both inflammatory and cytoskeletal pathways. In 21 dps monolayers, particularly, we detected variations of its intracellular localization associated with the inflammatory stimulus and its mRNA/protein increase associated with the differentiated 21 dps enterocyte-like monolayer compared to the undifferentiated cells.

## 1. Introduction

The gastrointestinal (GI) absorptive epithelium is a functional barrier that controls fluxes of solutes and nutrients, as well as antigen identification and surveillance by mucosal immune cells [1,2]. The epithelium plays an important role in inflammation by serving as an interface between invading pathogens and the immune system of the host. During inflammation, the protective mechanisms are compromised by pathological processes affecting the barrier function; these include increased epithelial permeability consequent to mucosal inflammation as found in several human diseases, including inflammatory bowel disease (IBD), celiac disease, and irritable bowel syndrome (IBS) [2]. A direct target of these processes is the enterocyte monolayer, whose physiological function is early affected by inflammation-induced onsets acting on cytoskeletal dynamics. The reorganization of enterocyte apical junctions mediates epithelial barrier dysfunction, and in this context, the actin cytoskeleton plays an important role in regulating junctional integrity and remodeling under physiological and pathological states [3].

Many studies have described the effects of inflammatory triggers on the organization of the monolayer’s cytoskeleton along the GI. In patients with Crohn’s disease, induction of cell stress alters the cytoskeleton in intestinal epithelial cells via dysregulation of actin-binding proteins (e.g., villin 1 and gelsolin) [4]. Moreover, proteomic analyses of intestinal tissues obtained from animal models of colitis and IBD patients have shown that mucosal inflammation in vivo induces significant changes in the expression of epithelial actin itself and a number of actin-binding proteins [5]. In addition, cytoskeletal reorganization has been well studied in vitro in cell models of intestinal epithelial monolayers exposed to different inflammatory stimuli. For example, treatment of intestinal epithelial cells (IECs) with IFN-γ modifies the peri-junctional F-actin cytoskeleton by marked depolymerization [6] or development of subapical F-actin-coated vacuoles [7]. Similar effects have been observed after IECs exposure to ROS [8]. The regulation of cytoskeletal actin is closely associated with proteins that control different steps of the dynamic remodeling of the actin cytoskeleton, such as nucleation, elongation, and depolymerization [9].

Allograft inflammatory factor 1 (AIF-1), also known as ionized calcium-binding adaptor molecule 1 (IBA-1), is a protein with EF-hand calcium-binding domain expressed in microglia, monocytes, macrophages, and lymphocytes [10,11]. AIF-1 shows actin-cross-linking activity and is involved in the rearrangement of the actin cytoskeleton, such as membrane ruffling and phagocytosis [12,13,14]. On the other hand, AIF-1 plays a central role in inflammation by affecting the production of cytokines, expression of adhesion molecules, and inflammatory mediators [15,16]. Indeed, increased expression of AIF-1 has been observed in several autoimmune and inflammatory diseases, including rheumatoid arthritis, diabetes, and neurodegenerative disorders. Recently, the involvement of AIF-1 in the regulation of intestinal inflammation has also been investigated. Kishikawa et al. [17] have focused on AIF-1s role in M cells, specialized antigen-sampling cells that take up intestinal luminal antigens, showing that it is critical for the transcytosis of bacteria. However, little is known about the function of AIF-1 in intestinal epithelial cells.

Among many cell models, the Caco-2 cell line from human colorectal adenocarcinoma is an established system for investigating in vitro GI absorption, epithelial permeability, and morpho-functional responses to physiological and pathophysiological stimuli [18,19]. Caco-2 cells are able to undergo spontaneous differentiation over time, switching their morphological phenotype from the tumor-like to the enterocyte-like phenotype functionally mimicking the absorptive monolayer of the small intestine, developing microvilli, tight junctions and expressing brush border associated enzymes [20,21].

Extensive studies have demonstrated that amino acids may take part in anti-inflammatory responses and in the regulation of intestinal epithelial barrier functions by maintaining or restoring IEC homeostasis [22]. In Caco-2 cells, glutamine has been found to be involved in the regulation of transepithelial resistance and in the expression of tight junction proteins [23]. Histidine has been found to inhibit hydrogen peroxide- and TNF-α-induced IL-8 secretion in both Caco-2 and HT-29 cells [24]. Recently, both synthetic and natural small peptides have been considered for their promising anti-inflammatory activities and are nowadays studied as therapeutic agents for a broad range of illnesses, from rheumatoid arthritis to Alzheimer’s and gut diseases [25,26]. Among small peptides, carnosine (β-alanyl-L-histidine, CAR) is a dipeptide present in excitable tissues as deriving from both/either endogenous synthesis and/or dietary supplementation [27]. CAR has a physiological role in maintaining cellular homeostasis, acting, e.g., as a free radical scavenger against oxidative stress [28], divalent metal ion chelator [29], or as a quencher of reactive carbonyl species [30]. The literature on CAR also includes data in relation to the GI tract mainly due to its presence as a component of animal food that may activate brain-gut interaction pathways. Many experimental studies report the ability of CAR to activate intestinal epithelial cells and induce the secretion of factors that activate brain function [31,32,33]. The protective effects of CAR in the case of GI inflammation [34] and its regulatory impact on cytokine secretion [35] have been studied. However, evidence of possible CAR actions on cytoskeleton remodeling in intestinal epithelium under inflammatory conditions has scarcely been reported.

Here, we evaluate changes in the actin cytoskeleton in Caco-2 cells forming monolayers at two different stages of spontaneous differentiation, i.e., undifferentiated tumor-like cells at 7 dps (days post-seeding) and differentiated enterocyte-like cells at 21 dps. Caco-2 monolayers were challenged by phorbol-12-myristate-13-acetate (PMA), adopted as in vitro proinflammatory stimulus on GI epithelial cells [36,37]. PMA effects were also evaluated in the presence of CAR, which revealed its ability to counteract early PMA-induced alterations on cytoskeletal actin remodeling, specifically in mature enterocyte-like monolayers. In such experimental settings, being AIF-1 a key activator of inflammatory mechanisms as well as a regulator of the intracellular actin dynamics involved in cytoskeletal responses in immune cells during inflammation [12,15], our aim is to find out its expression features with the hypothesis that they are associated with cytoskeletal actin modifications in enterocyte-like cells stimulated by inflammatory insults. Here, we report for the first time the molecular identification of AIF-1 expression in a GI epithelial monolayer model exposed to an inflammatory agent, showing its intracellular localization and mRNA/protein variations while marking different stages of the monolayer’s maturation in vitro.

## 2. Materials and Methods

### 2.1. Reagents and Materials

All chemicals, reagents, and kits were purchased at cell culture/molecular biology grade. Plasticware was invariably purchased, sterilized, disposed of, and treated for cell culture. Fetal bovine serum (FBS), Dulbecco’s phosphate buffer saline (D-PBS), Eagle’s minimum essential medium (MEM), penicillin/streptomycin solutions, trypsin, L-glutamine and non-essential amino acids were purchased from Corning-Fisher Scientific (Rodano, Italy). The compound 4′,6-Diamidino-2-Phenylindole (DAPI; Cat.: 28718-90-3), Triton X-100, phorbol 12-myristate 12-acetate (PMA; Cat. 16561-29-8), paraformaldehyde (PFA; Cat. 30525-89-4), bovine serum albumin (BSA) were obtained from Sigma-Aldrich (Milano, Italy). Phalloidin TRITC-conjugated (Cat.: R415) and L-carnosine (Cat. 305-84-0) were purchased from Thermo Fisher Scientific (Monza, Italy).

### 2.2. Cell Culture and Treatments

Human epithelial Caco-2 cells (ATCC n. HTB-37™) were grown at 37 °C, in a humidified atmosphere (5% CO_2_ in air), in MEM supplemented with 10% (*v*/*v*) FBS, 2 mM L-glutamine, 100 µg/mL penicillin-streptomycin and 1% (*v*/*v*) non-essential amino acid mix solution. The culture medium was replaced every third day, and propagation occurred routinely every 4–5 days post-seeding; all the experiments were conducted between passages 3 and 10 of propagation. For the experimental treatments, Caco-2 cells were used after continuous growth for 7 days post-seeding (dps) in standard culture conditions or for 21 dps to obtain a spontaneously differentiated intestinal epithelial monolayer (“enterocyte-like”) according to standard Caco-2 cell differentiation protocols [19,20].

Before the treatments assay, Caco-2 cells were seeded in 12-well plates (Corning-Fisher Scientific, Rodano, Italy) at a density of 0.5 × 10^5^ cells per well and then cultured for 7 and 21 dps under the standard conditions described above. At both 7 dps and 21 dps, Caco-2 monolayers were incubated for 48 h in the presence of 150 nM PMA, 150 nM PMA + 1 mM carnosine (CAR), or 1 mM CAR alone (final concentrations added to the culture medium).

### 2.3. Fluorescence Imaging of Cytoskeleton/Nuclei and Quantitative Image Analyses

For the analysis of cytoskeletal morphology, 1.5 × 10^5^ cells per well were seeded and grown on previously autoclaved, U.V.-sterilized glass coverslips leaned to the bottom of 6-well plates. After the previously described treatments, cells were washed thrice with D-PBS and fixed with 4% (*w*/*v*) PFA in D-PBS for 60 min at room temperature. Fixed samples were washed again thrice with D-PBS and then permeabilized with 0.1% (*w*/*v*) Triton X-100 in D-PBS for 20 min at room temperature. For staining the actin cytoskeleton, cells were incubated with 1 µg/mL phalloidin-TRITC for 20 min in the dark and washed three times with D-PBS. Coverslips were reverse mounted on glass slides with the Vectashield^®^ mounting medium containing DAPI (1.5 µg/mL) for nuclear staining. Fluorescence imaging of Caco-2 cells in monolayers was performed with a Nikon Eclipse 800 microscope equipped with the Nikon-NIS-elements-D package suite software (Version 3.07).

In phalloidin-TRITC and phalloidin-TRITC/DAPI merged images, Caco-2 morphometric size analyses were carried out using the ImageJ open-source image processing program (https://imagej.net/ImageJ, accessed on 27 December 2021); cell number was manually counted using the manual cell counting system offered by the ImageJ software in the selected region of interest (450 × 600 µm area). For each calculation, four representative areas were selected from each of the three biological replicates. The cell size was expressed by measuring a) the maximum cell diameter (obtained by joining the farthest poles of a single cell) and b) the area comprised in the cell perimeter traced around the marked circumferential actin belt. The fluorescence intensity of the actin cytoskeleton was calculated as integrated density (area × mean gray values) in the selected areas covering the cell perimeter. If not otherwise stated, average cell counts, diameter lengths, areas, and fluorescence intensities were expressed as a percentage with respect to the untreated control cells at 7 dps (100%).

### 2.4. Immunocytoichemistry

Caco-2 cells (1.5 × 10^5^) were seeded and grown on autoclaved, U.V.-sterilized glass coverslips for 7 and 21 dps, then fixed with 4% (*w*/*v*) paraformaldehyde (PFA) after experimental treatments as described above. Cells were washed with PBS (3 times for 10 min), permeabilized with 0.1% (*w*/*v*) Triton X-100 in PBS for 10 min at room temperature, pre-incubated for 40 min with PBS containing 1% (*w*/*v*) bovine serum albumin (BSA) and 1% (*v*/*v*) normal donkey serum, then incubated overnight at 4 °C with primary rabbit polyclonal anti-AIF-1 (Proteogenix, Schiltigheim, France) (dilution 1:5000). The washed specimens were incubated for 1 h at room temperature with the goat anti-rabbit fluorescein isothiocyanate (FITC)-conjugated (l_ex_ 493 nm, l_em_ 518 nm) secondary antibody diluted 1:2000 (Abcam, Cambridge, UK). Nuclei were counterstained by incubating cells for 5 min with 4,6-diamidino-2-phenylindole (DAPI, 0.1 mg/mL in PBS, lex 340 nm, lem 488 nm). Coverslips were mounted with 50% glycerol on glass slides and examined with an LSM 700 confocal laser microscope (Zeiss, Dresden, Germany), Zen2012 Black Edition program.

The quantification of AIF-1 immunofluorescence was carried out using the ImageJ open-source image processing program (https://imagej.net/ImageJ, accessed on 27 December 2021). In each image, ten digital areas were selected in appropriate regions, and the corresponding fluorescence intensity was calculated as integrated density (product of area and mean gray value). For each of the three biological replicates, the average values of fluorescence intensity were expressed as a percentage with respect to the untreated control cells at 7 dps (100%).

### 2.5. Total RNA and Protein Extraction

Simultaneous RNA and protein extractions from cell cultures were performed using the All-Prep DNA/RNA/Protein mini kit (Qiagen, Hilden, Germany) according to the manufacturer’s instructions. Briefly, cells grown in multi-well plates were washed twice with D-PBS and then lysed with the kit lysis buffer by scraping directly on the plate surface. At the end of the RNA/protein extraction protocols, RNA aliquots were stored in RNase-free conditions at −80 °C until use. RNA concentrations were calculated by spectrophotometry with the NanoDrop ND-2000 Spectrophotometer (Nanodrop Technologies, Wilmington, DE, USA), and the λ_260_/λ_280_ ratios were calculated to evaluate RNA purity; all the RNA extractions were qualitatively tested by electrophoresis of RNA samples on 1% (*w*/*v*) agarose gels. Protein concentrations in extracts were calculated by the Bio-Rad Protein Assay Dye Reagent Concentrate, according to the manufacturer’s protocol.

### 2.6. Primer Design and Real Time PCR (qPCR) Assays

The mRNA reference sequences of the investigated genes were collected from the GenBank database (https://www.ncbi.nlm.nih.gov/gene, accessed on 29 October 2022) and were used to select oligonucleotide sequences as primer pairs for consequent real-time PCR (qPCR) assays. By mRNA-to-genomic sequence alignment, the gene-specific forward and reverse primers were designed on different exons (intron spanning) to avoid amplification of genomic DNA. The AmplifX software version 2.0.7 was used to test PCR size, GC content, end stability, self/cross-dimer formation, and melting temperature for the selected primer pairs. Details of the gene-specific oligonucleotide sequences are reported in Table 1. Reverse transcription on the extracted RNAs was performed on 700 ng total RNA for each sample, using the iScript Select cDNA Synthesis kit (Bio-Rad, Segrate, Italy) according to the manufacturer’s instructions, with random primers in the reaction mix. Before qPCR analysis, primer pairs were tested for efficiency according to the amplification efficiency parameters for genes of interest and internal controls proposed by Schmittgen and Livak [38]. qPCR assays were performed using the iTaq Universal SYBR Green Supermix (Bio-Rad) with a CFX96 Touch™ Real-Time PCR Detection System (Bio-Rad). In the qPCR analysis, gene expression relative quantification was performed by analyzing the threshold values (CT) with the comparative CT method (also referred to as the 2^−ΔCT^ or 2^−ΔΔCT^ method), and qPCR data shown were the 2^−ΔCT^ values, which are considered as proportional to the amount of detected target mRNA. For each target gene and internal control (housekeeping), ΔCT values (ΔCT = target gene CT—housekeeping gene CT) were obtained from 2 different rounds of qPCR for each of the three biological replicates. Statistical analysis was performed after the 2^−ΔCT^ transformation [38].

### 2.7. Western Blotting

Following electrophoresis on stain-free pre-cast 12% polyacrylamide gels (Bio-Rad), proteins from cell lysates were electro-transferred to polyvinylidene fluoride (PVDF) membranes. After protein transfer, the membranes were blocked for 5 min in Every Blot Blocking Buffer (Bio-Rad) and then immunoblotted in Every Blot Blocking Buffer with the following primary antibody: mouse monoclonal anti-β-actin (Sigma-Aldrich, Milano, Italy; clone AC-15, product ID A5441; dilution 1:20,000) and rabbit polyclonal anti-AIF1 (Proteogenix, product ID A002181; dilution 1:1000). After rinsing, blots were incubated with conjugated anti-mouse/anti-rabbit secondary antibody (Sigma-Aldrich) and the immune reactive bands were detected using an enhanced chemiluminescence method (ECL kit, Bio-Rad) with ChemiDoc^TM^ Imaging System (Bio-Rad). Densitometric quantitative analysis was carried out using the Image Lab software version 6.1 (Bio-Rad). The pixel intensity for each band was analyzed, the background was subtracted, and the protein expression results were normalized with respect to the total lane of the loaded samples. As molecular weight markers, the Precision Plus Protein All Blue Standards (Bio-Rad, cat. no. #161-0373) were used.

### 2.8. Statistical Analysis

Unless otherwise stated, all data were expressed as means ± standard error of the mean (S.E.M.). Data means derive from two independent assays for each of the three biological replicates. Statistical analysis by two-tailed unpaired Student’s *t*-test or one-way ANOVA followed by Dunnett’s multiple comparison test was performed using GraphPad Prism 9.4.0; *p*-value ≤ 0.05 were considered significantly different.

## 3. Results

### 3.1. Morphological Features of Actin Cytoskeleton in Caco-2 Cell Monolayers Grown up to 7 and 21 Days Post Seeding (dps)

As shown by actin cytoskeleton/nuclei staining, in the absence of treatments, Caco-2 cells grown for 7 dps exhibit morphological features of an undifferentiated monolayer, i.e., discontinuous growth and a tendency to create proliferation gaps on the growth surface. Nuclei are different in size, with the occurrence of irregular shapes. The actin elements appear heterogeneously marked; some monolayer patches show regular cell-cell contact and adhesion structures, while in other optical fields, the cytoskeletal actin appears less organized with less defined rings in the cell-cell contact areas (Figure 1a). On the contrary, the 21 dps monolayer shows a highly regular mosaic-like organization of differentiated enterocyte-like cells which are small in size and show a global morphological homogeneity compared to the cells in the 7 dps monolayer. At 21 dps, gaps, discontinuities, and irregular nuclei are scarcely present. The homogeneous organization of the cytoskeleton is characterized by detectable cell-cell contact and adhesion structures with highly defined borders; the peripheral actin thickenings depict an epithelial organization with physiologically appropriate tightness (Figure 1b).

### 3.2. Effects of PMA Treatments on Morphology, Morphometry and Actin Cytoskeleton of Cells in Monolayers at 7 and 21 dps

Caco-2 monolayers at 7 and 21 dps were treated for 48 h with proinflammatory 150 nM PMA or 150 nM PMA + 1 mM CAR. At 7 dps, the treatments induced no differences in the monolayer cell density. The overall number of cells remained stable compared to the control, although a faint reduction (but without statistical significance, *p* > 0.05) can be noticed in the case of both treatments, PMA and PMA+CAR (Table 2). At 21 dps, in untreated monolayers, cell counts show a significant increase in cell number with respect to 7 dps untreated cells (244.8% ± 20.2 vs. 100% ± 6.1, respectively; *p* < 0.05). In addition, at 21 dps, the cell counts indicate that the 21 dps monolayer is strongly sensitive to PMA in terms of reduction in the cell population (35.7% ± 7.8 vs. 100% ± 8.2 of the respective 21 dps untreated control; Table 2), and that CAR added to PMA-treated cells tendentially counteracts the reduction (44.7% ± 4.2 vs. 100% ± 8.2 of the respective untreated control; Table 2). In particular, these PMA-induced variations in cell counts are in agreement with cell viability data under the same treatment conditions (see MTT test in Appendix A).

Cell diameters and areas were also measured. At 7 dps, Caco-2 cells grown in a monolayer showed no significant changes in the diameter or area after the experimental treatments with respect to the control cells (Table 2). At 21 dps, untreated cells in monolayer showed significant reductions in diameter and individual area, compared to untreated cells at 7 dps (diameter: 48.8% ± 1.3 vs. 100% ± 5.9, respectively, *p* < 0.001; area: 41.7% ± 1.8 vs. 100% ± 8.8, respectively, *p* < 0.05). At 21 dps, PMA-affected cell diameter was found to increase vs. the respective untreated control (230.3% ± 12.3 vs. 100% ± 2.7, respectively, *p* < 0.001); likewise, the PMA-affected area of 21 dps cells resulted in a significant increase (386.5% ± 62.9 vs. 100% ± 4.3, respectively, *p* < 0.05) compared to the control (Table 2). In mature monolayers at 21 dps, CAR partially contained the PMA-induced alterations in terms of the area measured for individual cells (Table 2).

The fluorescence intensity of cytoskeletal actin in cells was also quantitatively evaluated (Table 2). At 21 dps, untreated cells in monolayer showed a significant reduction in normalized fluorescence compared to untreated cells at 7 dps (i.e., 55.6% ± 4.6 vs. 100% ± 11.5, respectively, with *p* < 0.05). At 7 dps, fluorescence intensity from cytoskeletal actin elements is not significantly affected by treatments; contrariwise, at 21 dps, it is significantly increased in PMA-treated monolayers compared to control (232.4% ± 19.1 vs. 100% ± 8.2, respectively, *p* < 0.01); moreover, at 21 dps presence of CAR avoids the PMA-induced increasing.

Through fluorescence imaging, qualitative effects of PMA on cytoskeletal actin organization were also assessed. In undifferentiated 7 dps Caco-2 cells, PMA causes minor changes in the organization of F-actin fibers with respect to control; however, some alternations of frayed and thickened structures were noticed. As in the controls (Figure 2a), cytoskeletal actin elements of PMA-treated cells are overall irregular and enlarged, resulting in different cell shapes and sizes. Moreover, a subpopulation of nuclei exhibits morphological alterations (Figure 2b).

At 21 dps, PMA-induced actin remodeling in differentiated enterocyte-like cells was also analyzed. Compared to the controls (Figure 2d), intensified intracellular stress fibers and punctuations in cell-cell contact areas appear, while thinner actin rings identify the borders of the apical domain of the cells. Consequently, the rearrangement of adjacent cells is altered, showing discontinuity in cell-cell borders. In contrast to the highly regular 2D architecture of the 21 dps control monolayer, the mosaic-like organization undergoes significant alterations: cells lose their regular polygonal geometry, becoming enlarged with partly roundish shape and variable size (Figure 2e). Remarkably, both at 7 dps and 21 dps stages, administration of CAR concurrent with PMA showed counteractive effects: specifically, at 7 dps, CAR preserves cytoskeletal actin organization keeping it comparable to the control (Figure 2c); at 21 dps, actin rings and cytoskeleton elements in cell-cell contact areas show fewer alterations than what observed in monolayers treated with PMA alone (Figure 2f).

The responsiveness of the actin cytoskeleton was also evaluated by Western Blot analysis of the β-actin (ACTB) protein. In the untreated 21 dps mature enterocyte-like monolayer, β-actin basal levels appeared higher compared to untreated cells at 7 dps (499.9% ± 230.9 vs. 100% ± 35.1, respectively). Both in 7 and 21 dps monolayers, β-actin protein levels showed an up-regulation trend after PMA treatments compared to the respective control, although without statistical significance (Figure 3).

### 3.3. Identification of the Allograft Inflammatory Factor 1 (AIF-1) Gene Products in 7 and 21 dps Caco-2 Monolayers Exposed to PMA Challenge

Expression of AIF-1 (i.e., allograft inflammatory factor 1, actin-binding protein) gene products was evaluated in 7 and 21 dps Caco-2 monolayers in the absence of treatments and after 48 h exposure to PMA. AIF-1 mRNA expression was detected in untreated control cells both in 7 dps and 21 dps monolayers and, interestingly, its expression significantly increased in mature monolayer (21 dps) vs. 7 dps (140.6% ± 5.6 vs. 100% ± 12.9 with *p* < 0.05, Figure 4a); as the same, Western Blot analysis showed protein increase at 21 dps vs. 7 dps (213.3% ± 41.8 vs. 100% ± 30.2, respectively. Figure 4b). When cells were exposed to PMA or PMA+CAR, no statistically significant changes in AIF-1 protein were detected. However, faintly decreasing trends in both the 7 dps (undifferentiated) and the 21 dps mature monolayer are observed when compared to the respective controls (see table in Figure 4b).

Once the expression of AIF-1 gene products was identified, intracellular localization of AIF-1 protein was established in monolayers at both 7 and 21 dps differentiation stages. Fluorescent immunocytochemistry indicated that in 7 dps control cells, AIF-1 protein is distributed in small intracellular vesicles localized closely to perinuclear domains (Figure 5a). After PMA treatments at 7 dps, the AIF-1 immunoreactive pattern was found to be conserved in control, but the intensity and number of AIF-1 positive inclusions appeared to decrease and were distributed in clusters of cells in the monolayer (Figure 5b). Quantitative analysis of the fluorescence intensity indicated that AIF-1 fluorescence was significantly reduced in 7 dps monolayers after both PMA and PMA+CAR treatments with respect to the control (~0.6 fold decrease for both treatments with *p* < 0.0001; see graph in Figure 5g).

At 21 dps, control cells in the monolayer displayed a homogeneous cytoplasmic diffused distribution of AIF-1 signal in association with some positive vesicles observed in the perinuclear area (Figure 5d). In 21 dps mature monolayers, PMA treatment resulted in an intensification of AIF-1 immunoreactivity in vesicle-like formations, variably sized, in which intense signals were frequently found in perinuclear domains (Figure 5e); in the presence of CAR, these AIF-1 positive inclusions near cell nuclei were reduced, while cytoplasmatic immunoreactivity remained relatively similar to PMA-treated cells (Figure 5f).

Compared to 7 dps, quantitative analysis of fluorescence intensity in 21 dps mature monolayers showed that the AIF-1 signal is only slightly reduced after PMA and PMA+CAR exposure vs. its respective control (~0.1 negative fold change for both treatments, with *p* < 0.05; Figure 5g).

### 3.4. Cytoskeleton-/Inflammation-Related mRNA Expression Variations Induced by PMA Treatments in 7 and 21 dps Caco-2 Monolayers

The mRNA expression of some genes related to cytoskeletal dynamics (FAK, focal adhesion kinase; ITGB1 integrin) and inflammatory response (NFKB1, nuclear factor kappa B1; SCL15A4/PHT1, peptide/histidine transporter) was investigated by qPCR assays in cells at 7 dps and 21 dps exposed to PMA and PMA+CAR for 48 h (see Figure 6).

The expression levels of FAK mRNA in 7 dps Caco-2 cells, as well as in mature monolayers at 21 dps, did not significantly change by all experimental treatments. When ITGB1 mRNA levels were analyzed in 7 dps undifferentiated monolayers, no changes occurred upon treatments; on the other hand, in 21 dps enterocyte-like monolayers, ITGB1 mRNA significantly increased 6.5- and 13.6-fold by PMA and PMA+CAR, respectively, compared to their respective control. Interestingly, for both FAK and ITGB1 mRNAs, basal levels in control monolayers appear reduced at 21 dps vs. 7 dps.

The NFKB1 mRNA was not significantly modulated by the differentiation stage and/or treatments. Contrarily, when the mRNA levels of the SLC15A4/PHT1 gene were analyzed, SLC15A4/PHT1 mRNA was only faintly modulated by PMA treatments in 7 dps monolayers, while a strong, statistically significant PMA-induced up-regulation was detected in 21 dps enterocyte-like monolayers (~8-fold increase); moreover, the concomitant presence of CAR provoked avoidance of SLC15A4/PHT1 mRNA variations, at 21 as well as 7 dps monolayers’ differentiation stages.

## 4. Discussion

Understanding the mechanisms of regulation of the intestinal epithelial barrier both in physiological and inflammatory conditions is fundamental to ensure the maintenance of intestinal homeostasis, as well as it is crucial to clarify the direct role of the intestinal epithelium itself in terms of immune and anti-inflammatory responses.

The purpose of this work has been the analysis of stage-specific responses in undifferentiated and differentiated intestinal epithelial cells challenged by the action of an inflammation-mimicking agent, such as PMA, in order to exploit, at best, a consistent in vitro model of enterocyte-like cells and their pathophysiological behavior during inflammation. More in detail, experiments with PMA have been focused on the association of the monolayer’s morphology and/or rearrangements of cytoskeletal actin with a proinflammatory stimulus. Moreover, the presence of a physiological trigger such as the carnosine dipeptide (CAR), which has been studied for multifunctional protective effects, in the GI context also [27,28,29,30,31,32,33,34], has been considered. In this context, the human Caco-2 cells proved to be optimal, based on their ability to spontaneously differentiate starting from their “tumor-like” state, i.e., at 7 dps, to the mature monolayer at 21 dps exhibiting morpho-functional properties of small intestine enterocytes [19,20].

As shown by our results, 48 h of incubation with PMA induces reorganization of the cytoskeletal actin structures both in 7 and 21 dps Caco-2 cells in monolayer. Nevertheless, fluorescence images of the cytoskeleton in 21 dps monolayers show that actin organization is much more affected by PMA in terms of thinning and fragmentation of actin rings, interruption of cell-cell contact areas, and development of thickened punctuations and stress fibers. Interestingly, this evidence is also found in association with the variations of ACTB protein levels. This responsiveness to PMA higher at 21 dps than at 7 dps, in terms of cytoskeletal actin elements, is remarkably evident based on fluorescence and morphometric values (Table 2), indicating significant dimensional alterations of cells in monolayer, associated with a significant reduction in cell number. These findings agree with previous studies in which PMA treatment exerted cytoskeletal altering effects on different tissues and cell types via protein kinase C (PKC) family members activation [39], which regulates actin cytoskeleton and membrane dynamics [40,41]. Moreover, in vitro studies with Caco-2 and other epithelial cell lines demonstrated that acute activations (e.g., of PKC) induced by PMA lead to the formation of ventral actin structures and the redistribution of F-actin towards cell borders [37,42].

It is worth noting that PMA-induced rearrangements of actin cytoskeletal elements in mature monolayers are attenuated by the administration of CAR. While at 7 dps, the monolayer seems to be less sensitive to treatments, at 21 dps, the presence of CAR concurrent with PMA counteracts the increasing cell size and actin fluorescence intensity (Table 2). Moreover, CAR preserves actin organization, cell-cell contact areas, and cell shape compared to the control (Figure 2f). These observations suggest that CAR potentially ameliorates the effects on the actin cytoskeleton and cell morphology, thus helping with the maintenance of the monolayer’s homeostasis. Notably, treatments with CAR alone never elicited statistically significant differences compared to untreated control in all the experimental assays of our work. The protective properties of CAR have been demonstrated in various intestinal inflammatory processes. For example, in Caco-2 and HT29 intestinal cells, CAR increases the expression of tight junction proteins promoting epithelial barrier integrity [43]. The molecular mechanisms engaging CAR are not entirely clear, but it is suggested that one of its major targets is the mTOR signaling pathway involved in intestinal epithelium growth and proliferation. Supra-physiological doses of CAR are found to inactivate mTOR signaling in human gastric carcinoma cells [44]. Additionally, in Caco-2 cells, CAR inhibits the phosphorylation of eIF4E, resulting in the inhibition of the mTOR signaling cascade [35], in accordance with the hypothesis that this naturally occurring nutrient dipeptide is beneficial in the inflammatory response in mature intestinal epithelial cells.

Comprehensively, the PMA-induced actin cytoskeleton rearrangements are associated with morphological alterations of the cells in monolayers, but after the analysis of morphological features of untreated cells at both 7 and 21 dps stages, it is observed that PMA treatments do not alter cell morphology in the 7 dps undifferentiated monolayer; contrariwise, PMA dramatically modifies the architecture of the differentiated monolayer at 21 dps in which cells lose regular polygonal geometry. Taken together, these results suggest that the undifferentiated monolayer appears less sensitive to inflammation-mimicking stimuli; conversely, 21 dps mature monolayer better “feels” stimuli and reacts by remodeling geometry and cell size. As known, the actin cytoskeleton is a key determinant of animal cell morphology. Variations in the cellular actin network, in response to intrinsic and extrinsic factors, drive cell shape changes and determine cell fate during differentiation [45,46]. Considering the observed association between PMA treatments and alteration of monolayer architecture (cell size and density) with actin remodeling in enterocyte-like cells, it has been found that mature monolayer seems to show a physiological reaction, unlike the undifferentiated “tumor-like” monolayer, which exhibits a lack of responsiveness to inflammatory triggers. In this context, the experimental treatments adopted may be a cause of the activation of signaling pathways that control intestinal epithelium plasticity. Several in vivo and in vitro studies demonstrated that intestinal inflammation could trigger these pathways, i.e., Wnt, Notch, or YAP/TAZ signaling to restore homeostasis after colitis injury [46,47,48,49,50]. Interestingly, based on these experiments, it can be suggested that the mature epithelial monolayer model reacts to inflammatory insults through the dedifferentiation of enterocytes. In fact, disruption of the cell layer after PMA administration has not been observed, but treatments result in structural changes which cause loss of the differentiated enterocyte-like morphological phenotype. Challenged mature monolayers recover morphological and morphometric features similar to that seen in undifferentiated “tumor-like” monolayers (Table 2). Alterations in the differentiation status of intestinal epithelial cells associated with inflammation have already been observed under the action of other inflammatory mediators [49,51]. In vivo, Lyons et al. [52] demonstrated the link between differentiation and pro-inflammatory signaling of epithelium in a mouse model of colitis. This meets the hypothesis that, in our hands, Caco-2 cells become more sensitive to stimulation upon differentiation. In this study, the analyses of transcriptional expression by qPCR corroborate the evidence of PMA effects. Overall, trends of mRNA variations hint that mature enterocyte-like monolayers at 21 dps are physiologically responsive to PMA stimulation, whereas undifferentiated monolayer at 7 dps has not gained such responsiveness yet. More in detail, by treating cells with PMA, positive modulation of mRNAs functionally related to actin dynamics and cytoskeletal reactivity, such as ITGB1 and FAK, is associated with concomitant actin reorganization as observed in mature monolayer, according to the responsiveness of actin cytoskeleton network to the differentiation stage of Caco-2 cells. Interestingly, in mature 21 dps monolayers, basal levels for both FAK and ITGB1 mRNAs are reduced vs. 7 dps: in this view, PMA-dependent up-regulation enforces the idea that proinflammatory stimuli might switch cytoskeletal dynamics towards “dedifferentiation routes.”

From our results, great interest has come from the study of the AIF-1 gene products. AIF-1 (allograft inflammatory factor 1, aka IBA1) is an intracellular Ca^2+^-binding protein that plays a key role in the regulation of the intracellular actin dynamics through direct binding of actin, enhances membrane ruffling, and participates in phagocytosis and cell motility [12]. This protein promotes the activation of immune cells during inflammation [15]. Here, the first evidence for specific expression and localization of AIF-1 protein in enterocyte-like cells in vitro and possible variations induced by PMA has been provided. In this study, AIF-1 protein has been detected at both stages of epithelial monolayer maturation. At 7 dps, control cells show AIF-1 immunoreactivity in small vesicles near perinuclear domains. The PMA-challenged cells exhibit a reduction in protein levels, as also evidenced by quantitative fluorescence assessment and Western blot assays (Figure 4 and Figure 5g). In parallel, in 21 dps cells, AIF-1 gene products are present at higher levels compared to the 7 dps monolayer suggesting that this protein might putatively mark differentiation of epithelial monolayers (Figure 4a,b) and that, noteworthy, this might also be related to its involvement in actin dynamics as mentioned above [12]. Furthermore, mature monolayers show different immunocytochemical localization. In untreated 21 dps monolayers, AIF-1 is more diffusely located in the cytoplasm, while in PMA-treated cells, homogeneous cytoplasmic distribution appears associated with variably sized positive inclusions near nuclei. Changes in intracellular localization of AIF-1 after PMA treatment are fully in agreement with the described actin cytoskeleton rearrangements in the mature monolayer responding to PMA with F-actin and its cross-linking machinery reorganization. Despite PMA+CAR treatments did not elicit quantitative differences compared to PMA alone on AIF-1 mRNA/protein levels in our experiments, it is worth noting that CAR showed effects on the localization of the AIF-1-related immunoreactive vesicle-like inclusions in 21 dps enterocyte-like cells; in fact, CAR’s presence seemed to preserve the localization pattern seen in the untreated control, avoiding the perinuclear accumulation observed in cells treated with PMA alone. This evidence will need further investigation based on the intriguing hypothesis that the CAR molecule may act as a divalent ion chelator [27], thus interfering with intracellular pathways of AIF-1 as a Ca^2+^-binding protein [12]. Currently, the function of AIF-1 in Caco-2 cells and intestinal epithelia is unknown. To the best of our knowledge, this is the first identification of the protein in intestinal epithelial cells. Kishikawa et al. [17] reported AIF-1 expression in intestinal M cells, which participates in the activation of β1 integrin and actin remodeling. More recently, AIF-1 increase has been described in serum and tissue of rats and human colon mucosa during DSS-induced experimental colitis or Crohn’s disease, respectively [53,54]. In both species, AIF-1 is located just below the colonic epithelial cells and in immune cells infiltrating the lamina propria. Thus, the results presented here are surprising because they show the differential regulation of AIF-1 in GI epithelial cells and not only in mucosal immune cells during inflammation. From these data, AIF-1 emerges as a cytoplasmic protein in enterocytes and a putative marker of enterocyte maturation; on this basis, its characteristics and functional role in enterocytes deserve further investigation.

## 5. Conclusions

Our study highlights that the stage of differentiation of intestinal epithelial cell monolayer might be a determinant of response to exogenous inflammatory agents and/or physiologically active compounds. The undifferentiated monolayer (7 dps) shows no predictive responses under a PMA challenge. On the contrary, the enterocyte-like monolayer (21 dps) shows cytoskeletal rearrangements and gene expression variations in the epithelial layer, which are more evidently associated with (patho)physiological response to inflammatory stimuli. In this context, the dipeptide carnosine has been shown as potentially ameliorative against PMA in enterocyte-like monolayers but not in the undifferentiated ones, thus being in the former a potential aid for homeostasis balance. Such findings could aid in better exploiting the effects of both PMA and carnosine on Caco-2 monolayers cultured at different differentiation stages post-seeding. In addition to providing new data on PMA/carnosine treated/untreated cells, we also give hints of differential use and differential outputs based on the different levels of culturing over time. In this regard, some findings could be useful to discern biological responses dependent on the maturation stage of a Caco-2 monolayer from those not dependent.

Moreover, for the first time, we report the molecular identification and specific localization of AIF-1 in Caco-2 cells, untreated or treated with PMA, and in enterocyte-like cells in vitro. Such data suggest that this protein might be a putative marker in differentiating intestinal epithelial cells. To date, it has been only reported in intestinal M cells (which participate in the activation of β1 integrin and actin remodeling) but not in any enterocyte along the GI tract. On this basis, further study is warranted to obtain detailed knowledge of the AIF-1 role in intestinal epithelia in health and inflammatory disease.

## Figures and Tables

**Figure 1 biology-12-00036-f001:**
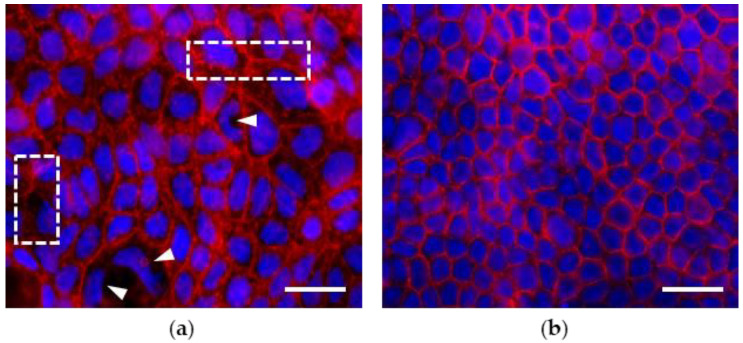
Morphology and organization of untreated Caco-2 monolayers. Cells were grown on coverslips for 7 (**a**) and 21 (**b**) days post-seeding (dps). At 7 dps (**a**), cells display different shapes and sizes and recurrence of amorphous nuclei (arrowheads), typical of undifferentiated tumor-like cells. Actin elements appear heterogeneously marked with evident vs. irregular cell-cell contact structures (rectangles). Contrariwise, 21 dps enterocyte-like cells (**b**) show regular polygonal geometry, small-size, and regular nuclei compared to the 7 dps monolayer (**a**). Circumferential actin belt is clearly visible, highlighting regular cell-cell-cytoskeletal junctions (**b**). Phalloidin-TRITC and Vectashield Mounting Medium with DAPI were used for the detection of actin elements (red) and nuclei (blue). Scale bar: 100 µm.

**Figure 2 biology-12-00036-f002:**
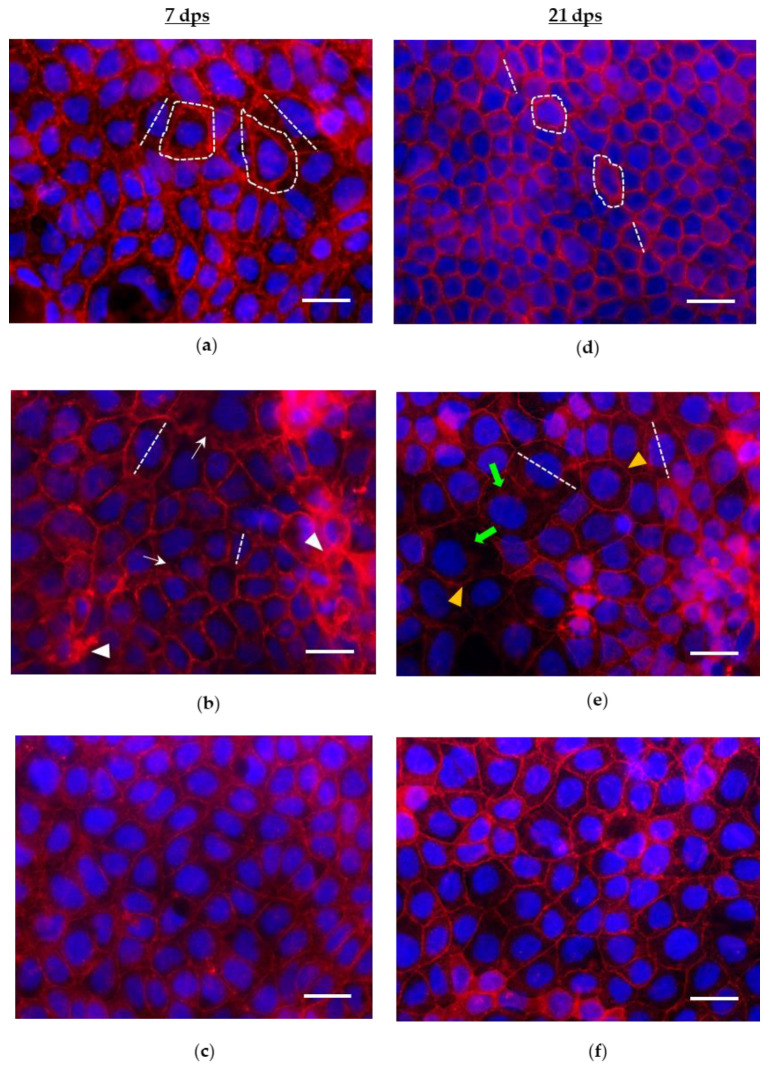
Phalloidin-TRITC/DAPI staining (merged images) of 7 and 21 dps Caco-2 monolayers treated with PMA (150 nM) or PMA+CAR (1 mM) for 48 h. (**a**–**c**): 7 dps monolayers; (**d**–**f**): 21 dps monolayers. (**b**): representative picture of 7 dps PMA-treated cells; white arrowheads and arrows indicate thickened and frayed F-actin rings, respectively; the monolayer architecture maintains the tumor-like organization as in the untreated control (**a**). (**e**): representative picture of 21 dps PMA-treated cells; actin rings appear thinned and discontinuous (yellow arrowheads); with respect to 21 dps untreated control (**d**), cells appear enlarged and irregularly roundish (green arrows). (**c,f**): in pictures both from 7 and 21 dps monolayers, concurrent exposure to PMA and CAR produces partial protection from the PMA-induced rearrangement/derangement of the actin cytoskeleton. In pictures, white dashed lines indicate representative cell diameters and perimeters designed to calculate the length, area, and fluorescence intensity of cells by applying ImageJ analysis software (see Materials and Methods for details). Unit areas: 450 × 600 µm; Scale bar: 100 µm.

**Figure 3 biology-12-00036-f003:**
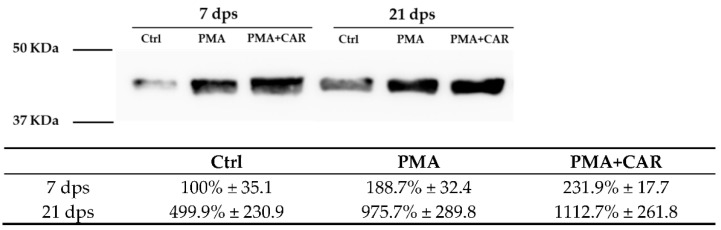
Expression of β-actin (ACTB) protein in Caco-2 monolayers exposed to PMA (150 nM) or PMA+CAR (1 mM). Western blot assays were performed using a specific antibody against the ACTB protein product (~42 kDa) on protein extracts from 7 dps and 21 dps Caco-2 monolayers in the absence (Ctrl) or presence of PMA and PMA+CAR for 48 h; molecular weight marker is shown on the left (Bio-Rad Precision Plus Protein All Blue Standard, cat. no. #161-0373). For quantitation in the table, densitometric values are normalized with respect to total lane protein amount (Bio-Rad Image Lab software version 6.1), then data are presented as mean ± SEM of n = 3 independent biological replicates expressed as a percent with respect to the control mean value at 7 dps (Ctrl = 100%). Statistical analysis: one-way ANOVA with Dunnett correction for multiple comparisons.

**Figure 4 biology-12-00036-f004:**
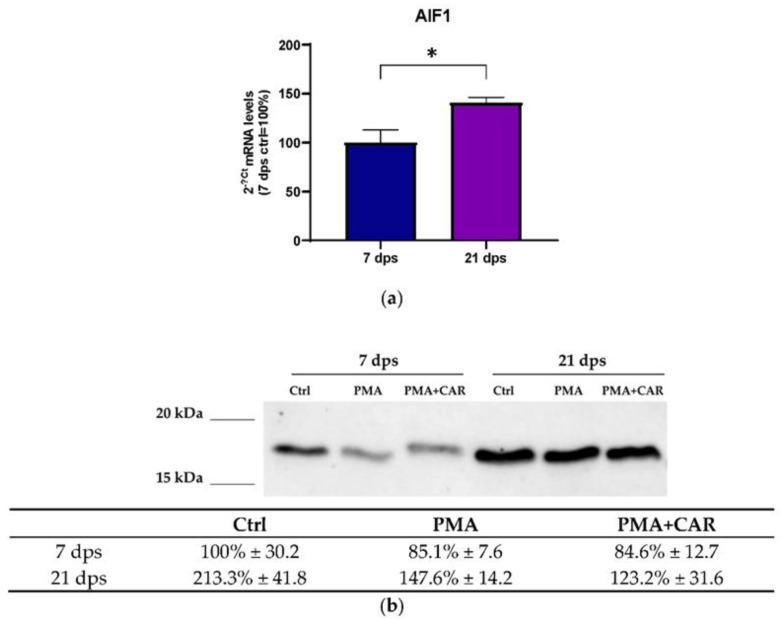
Expression of AIF-1 gene products in Caco-2 cells exposed to PMA (150 nM) and PMA+CAR (1 mM). (**a**) mRNA expression analysis by qPCR on total RNA extracted from 7 dps and 21 dps Caco-2 monolayers. Amounts of the target AIF-1 mRNA were calculated as 2^−ΔCt^ mean values obtained from two rounds of real-time PCR assays for each of three independent biological replicates (see Materials and Methods for details) and normalized with respect to the GAPDH mRNA (housekeeping). Values are expressed as a percent with respect to the control mean value at 7 dps (100%). Statistical analysis: two-tailed unpaired Student’s *t*-test (* *p* < 0.05). (**b**) Representative image of Western Blot assays performed using a specific antibody against the human AIF-1 protein product (~17 kDa) on protein extracts from 7 dps and 21 dps Caco-2 monolayer in the absence (Ctrl) or the presence of PMA and PMA+CAR for 48 h; molecular weight marker is shown on the left (Bio-Rad Precision Plus Protein All Blue Standard, cat. no. #161-0373). For quantitation, densitometric values are normalized with respect to total lane protein (Bio-Rad Image Lab software version 6.1). Data are presented as mean ± SEM of three independent biological replicates and then expressed as a percent with respect to the control mean value at 7 dps (Ctrl = 100%) in the table and graph. Statistical analysis: one-way ANOVA with Dunnett correction for multiple comparisons.

**Figure 5 biology-12-00036-f005:**
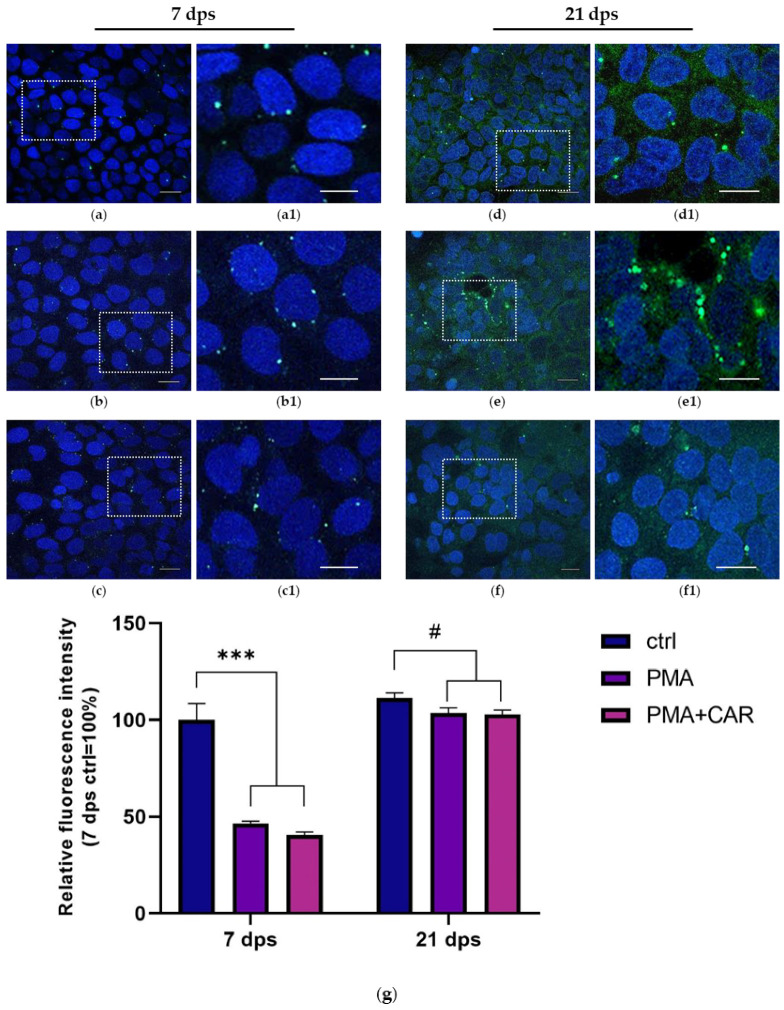
Fluorescence immunocytochemical analysis of AIF-1 protein localization in 7 and 21 dps Caco-2 monolayers treated with PMA (150 nM) or PMA+CAR (1 mM) for 48 h. Merged images show the anti-AIF-1 green-fluorescent signal and DAPI nuclear staining (blue). (**a**–**c**): undifferentiated Caco-2 monolayers at 7 dps, untreated (**a**), PMA-treated (**b**) and PMA+CAR-treated cells (**c**). (**d**–**f)**: differentiated, enterocyte-like Caco-2 monolayers at 21 dps, untreated (**d**), PMA-treated (**e**) and PMA+CAR-treated cells (**f**). Rectangles in pictures (**a**–**f**) are further magnified in (**a1**–**f1**), respectively; magnification: 63×. Scale bar: 20 µm (**a**–**f**), 10 µm (**a1**–**f1**). Imaging by an LSM 700 confocal laser microscope (Zeiss, Dresden, Germany), Zen2012 Black Edition program. (**g**): quantitative analysis of AIF-1 fluorescence intensity (see Materials and Methods for detail). Data are presented as mean ± SEM of five measures from each of three independent biological replicates. Mean values are expressed as a percent with respect to the untreated control at 7 dps (100%). Significant differences are presented by individual symbols. #: vs. control at 21 dps. Statistical analysis: one-way ANOVA with Dunnett correction for multiple comparisons (# *p* ≤ 0.05; *** *p* < 0.001).

**Figure 6 biology-12-00036-f006:**
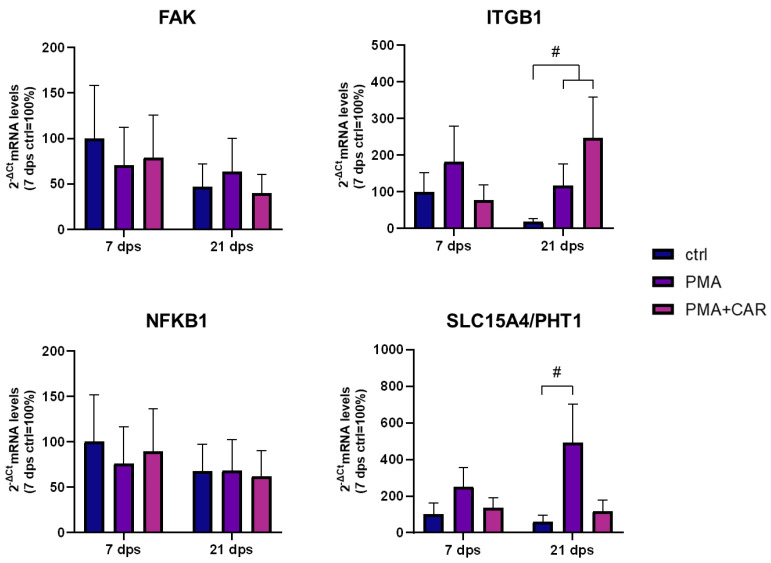
mRNA expression analysis by qPCR of cytoskeleton-/inflammation-related genes in 7 dps and 21 dps Caco-2 monolayer under PMA and PMA+CAR exposure for 48 h. Amounts of detected mRNAs (FAK, ITGB1, NFKB1, SLC15A4/PHT1) were calculated and expressed as 2^-ΔCt^ mean values obtained from two rounds of real-time PCR (qPCR) assays for each of three independent biological replicates (see Materials and Methods for details) and then were expressed as a percentage with respect to the control mean value at 7 dps (ctrl = 100%). Statistical analysis by one-way ANOVA with Dunnett correction for multiple comparisons (# *p* ≤ 0.05 vs. control at 21 dps).

**Table 1 biology-12-00036-t001:** Features of primer sequences for qPCR expression analysis**.** For each gene, the NCBI accession numbers of the mRNA reference sequence (RefSeq mRNA) used for primer design are reported. For each primer, the 5′-3′ nucleotide sequence and melting temperature (Tm) are reported. For each mRNA detection, the expected amplicon length is reported (PCR size) in base pairs (bp).

GENE	RefSeq mRNA Acc. No.	Sense Primer 5′-3′ (Tm)	Antisense Primer 5′-3′ (Tm)	PCR Size (bp)
*ACTB*	NM_001101.5	ATCGTGCGTGACATTAAGGAGA (56 °C)	TCCTCCCCTGGAGAAGAGCT (55 °C)	102
*FAK*	NM_001352694.2	ATTAAATGGATGGCTCCA (54 °C)	CTCCCACATACACACACC (58 °C)	121
*AIF1*	NM_001623.5	GAAGCGAATGCTGGAGAA (54 °C)	ATCTCTTGCCCAGCATCA (54 °C)	98
*ITGB1*	NM_002211.3	CAAATGCCAAATCATGTGGA (51 °C)	TTCTCTGCTGTTCCTTTGCT (54 °C)	226
*NFKB1b*	NM_003998.4	AATGCCTTCCGGCTGAGTC (59 °C)	AGGCTGCCTGGATCACTTCA (60 °C)	140
*SLC15A4*	NM_145648.4	TGAAGGCATTGGAGTCTTT (51 °C)	TGGAAATACACTGTCCAGTAA (51 °C)	168
*GAPDH*	NM_002046.7	AAACCTGCCAAGTATGATGA (51 °C)	TACTCCTTGGAGGCCATGT (54 °C)	217

**Table 2 biology-12-00036-t002:** Effects of PMA and PMA+CAR on morphometry values in 7 and 21 dps Caco-2 monolayers. Cell count and measures refer to area units = 450 × 600 µm. Maximum diameter, individual cell area, and actin fluorescence intensity are measured and normalized by ImageJ analysis software (see Materials and Methods for details). Data are presented as mean ± SEM of five measures from each of three independent biological replicates; for cell counts, absolute mean values (n) are reported with bold characters in round brackets. In bold black, mean values are expressed as percent respect to untreated control at 7 dps (100%); limited to 21 dps data, mean values are also reported in italics (in brackets) as percent respect to untreated control at 21 dps (100%); significant differences are represented by individual symbols; #: vs. control at 7 dps; *: vs. control at 21 dps. Statistical analysis: one-way ANOVA with Dunnett correction for multiple comparisons (^#^/* *p* < 0.05; ^##^/** *p* < 0.01; ^####^/**** *p* < 0.00001).

	Ctrl	PMA	PMA+CAR
		Cell count (%)	
7 dps	100 ± 6.1 (n = 24 ± 1)	86.5 ± 13.9 (n = 21 ± 3)	87.5 ± 4.5 (n = 21 ± 1)
21 dps	244.8 ± 20.2 ^##^ (n = 59 ± 5) (*100 ± 8.3*)	87.5 ± 19.2 (n = 21 ± 5) (*35.7 ± 7.8 **)	109.4 ± 10.3 (n = 26 ± 2) (*44.7 ± 4.2 **)
		Diameter max length (%)	
7 dps	100 ± 5.9	94.5 ± 4.8	100.7 ± 3.7
21 dps	48.8 ± 1.3 ^####^ (*100 ± 2.7*)	112.3 ± 6.0 (*230.3 ± 12.3 *****)	104.6 ± 6.3 (*214.4 ± 13.0 *****)
		Area (%)	
7 dps	100 ± 8.8	115.9 ± 17.5	108.5 ± 8.9
21 dps	41.7 ± 1.8 ^##^ (*100 ± 4.3*)	161.0 ± 26.2 (*386.5 ± 62.9 **)	94.8 ± 11.5 (*227.64 ± 27.5 **)
		Fluorescence intensity (%)	
7 dps	100 ± 11.5	115.4 ± 19.6	81.2 ± 4.5
21 dps	55.6 ± 4.6 ^#^ (*100 ± 8.2*)	129.2 ± 10.6 (*232.4 ± 19.1 ***)	40.6 ± 3.4 (*73.1 ± 6.1 **)

## Data Availability

Not applicable.

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
