# Peer review of "Cytoskeletal Responses and Aif-1 Expression in Caco-2 Monolayers Exposed to Phorbol-12-Myristate-13-Acetate and Carnosine"

_biology, 2022, doi:10.3390/biology12010036_

Round 1

Reviewer 1 Report

The study by Mazzei et al focusses on how “inflammatory triggers” influence the intestinal barrier of the gut.  This group uses CACO-2 cells as a model of the epithelial cells that line the intestine.  By exposing CACO-2 cells (both early polarization and late polarization) to two different inflammatory triggers (PMA and carnosine), this group attempts to understand how these triggers influence the organization of the actin cytoskeleton.  They observe that 48 hrs of incubation with PMA induces reorganization of the cytoskeletal actin structure both in early and late polarized Caco-2 cells and that carnosine counteracts this effect. The group also evaluates how allograft inflammatory factor 1 (AIF-1), a product that is known to be increased in several autoimmune and inflammatory diseases, is impacted by inflammatory triggers.  They also report, for the first time, the localization of AIF-1 in CACO-2 enterocytes.

The experiments are well-done and the data is presented in a logical manner.  A major drawback of the study is that they do not try to relate changes in the actin cytoskeleton that is induced by the inflammatory triggers to any sort of defect in barrier function.  Does pMA cause a change in the transepithelial resistance of the monolayer?  Experiments addressing this would help their study considerably.

Author Response

Authors’ Response:

We thank the reviewer for giving us the possibility to explain this issue. In this work, the direct evaluation of epithelial barrier functionality was out of our scope, in fact our experiments were conducted in monolayers grown on conventional culture surfaces and not in trans-well systems, which are the gold standard culture systems for any kind of functional testing of barrier function. On the other hand, we also could count on the fact that in the literature the detrimental effects of PMA (as a proinflammatory and/or cytotoxic agent) on the functionality/permeability of the epithelial barrier are well known and have been widely assessed in Caco-2 cells, also based on TEER measurements [e.g., see 1-5]. Thus, our investigation was targeted to specifically find out actin cytoskeleton dynamics and related morphometric changes of the Caco-2 monolayer exposed to PMA, apart from the implied connection to transepithelial permeability/resistance variations; also, as stated in the manuscript, we aimed at pointing our attention to the cytoskeletal actin in different maturation stages of Caco-2 monolayer, this being an aspect too often overlooked and lacking information in a large part of the literature based on Caco-2 cells as epithelial model in vitro. In parallel, we were interested in and aimed at obtaining the first molecular identification of AIF-1 protein expression in a GI epithelial cell model/monolayer; we wanted to obtain this “ab initio” information regardless of possible insights into permeability.

[1. Nowak P, et al. Int J Mol Med. 2004 Aug;14(2):175-8. PMID: 15254761]

[2. Turner JR, et al. Am J Physiol. 1999 Sep;277(3):C554-62. PMID: 10484342]

[3. Song JC, et al. Am J Physiol Cell Physiol. 2001 Aug;281(2):C649-61. PMID: 11443064]

[4. Li YH, et al. Acta Pharmacol Sin. 2014 Feb;35(2):283-91. PMID: 24362330]

[5. Sawai T, et al. Pediatr Surg Int. 2002 Oct;18(7):591-4. PMID: 12471472]

Reviewer 2 Report

This is an interesting study in which the authors investigated actin cytoskeletal changes in a Caco-2 cell monolayer model at different stages. In addition, they used Phorbol-12-myristate-13-acetata (PMA) to stimulate Caco2 monolayers to study the effect of inflammatory state on the gastrointestinal epithelium. In this paper, the authors found that differentiated monolayers (21dps) responded significantly to PMA stimulation, and therefore, they suggest that enterocyte-like monolayers (21dps) are a potential model for studying the (patho) physiological response of cells to inflammatory stimuli. In addition to this, the authors found that the addition of myostatin (CAR) attenuated the effect of PMA on the Caco2 monolayer. After all, the authors investigated the expression and localization of AIF1 in different stages of Caco2 monolayer and suggested that this could be a putative marker of differentiated intestinal epithelial cells.

However, this study has some limitations.

First, the hypothesis needs to be better formulated. The introduction section looks like an explanation of a few key words and does not give a clearer hypothesis and a link between AIF-1 and cytoskeletal responses.

Second, the authors only evaluated the Caco2 2D model with morphological features and actin expression, it would have been better to evaluate more metrics such as teer measurements and epithelial barrier markers.

Third, the authors found a significant decrease in cell number after treatment with PMA, and I would like to know if the authors tested the toxicity of PMA on cell viability. Also, the CRA only group was mentioned in the methods section, but not shown in the results section.

Minor comments, it would be better to use bar graphs to represent the results, which are easier for the reader to understand than tables.

Author Response

Comments and Suggestions for Authors

This is an interesting study in which the authors investigated actin cytoskeletal changes in a Caco-2 cell monolayer model at different stages. In addition, they used Phorbol-12-myristate-13-acetata (PMA) to stimulate Caco2 monolayers to study the effect of inflammatory state on the gastrointestinal epithelium. In this paper, the authors found that differentiated monolayers (21dps) responded significantly to PMA stimulation, and therefore, they suggest that enterocyte-like monolayers (21dps) are a potential model for studying the (patho) physiological response of cells to inflammatory stimuli. In addition to this, the authors found that the addition of myostatin (CAR) attenuated the effect of PMA on the Caco2 monolayer. After all, the authors investigated the expression and localization of AIF1 in different stages of Caco2 monolayer and suggested that this could be a putative marker of differentiated intestinal epithelial cells.

Authors’ response:

We are grateful to the reviewer for this general comment (and all the following), which gave us the actual possibility to ameliorate the manuscript. For the sake of precision, we recall that myostatin (reported in the reviewer's comment) was not used in this work but, instead, the use of the dipeptide carnosine is described; however we are fully confident that this is not an impediment to understanding our responses.

However, this study has some limitations.

  • First, the hypothesis needs to be better formulated. The introduction section looks like an explanation of a few key words and does not give a clearer hypothesis and a link between AIF-1 and cytoskeletal responses.

Authors’ response:

We thank the reviewer for her/his criticism. The hypothesis in the final part of the Introduction has been better described and changed accordingly.

  • Second, the authors only evaluated the Caco2 2D model with morphological features and actin expression, it would have been better to evaluate more metrics such as teer measurements and epithelial barrier markers.

Authors’ response:

We thank the reviewer for giving us the possibility to explain this issue. In this work, the direct evaluation of epithelial barrier functionality was out of our scope, in fact our experiments were conducted in monolayers grown on conventional culture surfaces and not in trans-well systems, which are the gold standard culture systems for any kind of functional testing of barrier function. On the other hand, we also could count on the fact that in the literature the detrimental effects of PMA (as a proinflammatory and/or cytotoxic agent) on the functionality/permeability of the epithelial barrier are well known and have been widely assessed in Caco-2 cells, also based on TEER measurements [e.g., see 1-5]. Thus, our investigation was targeted to specifically find out actin cytoskeleton dynamics and related morphometric changes of the Caco-2 monolayer exposed to PMA, apart from the implied connection to transepithelial permeability/resistance variations; also, as stated in the manuscript, we aimed at pointing our attention to the cytoskeletal actin in different maturation stages of Caco-2 monolayer, this being an aspect too often overlooked and lacking information in a large part of the literature based on Caco-2 cells as epithelial model in vitro. In parallel, we were interested in and aimed at obtaining the first molecular identification of AIF-1 protein expression in a GI epithelial cell model/monolayer; we wanted to obtain this “ab initio” information regardless of possible insights into permeability.

[1. Nowak P, et al. Int J Mol Med. 2004 Aug;14(2):175-8. PMID: 15254761]

[2. Turner JR, et al. Am J Physiol. 1999 Sep;277(3):C554-62. PMID: 10484342]

[3. Song JC, et al. Am J Physiol Cell Physiol. 2001 Aug;281(2):C649-61. PMID: 11443064]

[4. Li YH, et al. Acta Pharmacol Sin. 2014 Feb;35(2):283-91. PMID: 24362330]

[5. Sawai T, et al. Pediatr Surg Int. 2002 Oct;18(7):591-4. PMID: 12471472]

  • Third, the authors found a significant decrease in cell number after treatment with PMA, and I would like to know if the authors tested the toxicity of PMA on cell viability. Also, the CRA only group was mentioned in the methods section, but not shown in the results section.

Authors’ response:

We thank the reviewer for giving us the possibility to add a piece of information. We tested cell viability by MTT assays and we found significant viability reduction in agreement with the significantly decreased cell count at 21 dps. The graph related to such data has been added to a revised version of the supplementary material; moreover, the following sentence referring to this supplementary information has been added in the text (Results, paragraph 3.2): “In particular, these PMA-induced variations in cell counts are in agreement with cell viability data under the same treatment conditions (see MTT test in Supplementary material)”.

The group treated with CAR alone was an internal control group which did not exert any statistically significant difference compared to untreated control for all the experiments, thus we have excluded to show it for conciseness. Thanks to the reviewer’s comment, we have added the following specific sentence in the Discussion: “Notably, treatments with CAR alone never elicited statistically significant differences compared to untreated control, in all the experimental assays of our work”.

  • Minor comments, it would be better to use bar graphs to represent the results, which are easier for the reader to understand than tables.

Authors’ response:

We agree with the reviewer, nevertheless we have chosen this option for brevity and compactness, also for a more streamlined editorial format.

Reviewer 3 Report

In the manuscript titled ‘Cytoskeletal responses and AIF-1 expression in Caco-2 monolayers exposed to phorbol-12-myristate-13-acetate and carnosine’, the authors explored the cytoskeletal dynamic changes of enterocytes upon various treatments. The Caco-2 epithelial cell model was utilized in this study along with the inflammation-inducing stimulus (phorbol-12-myristate-13-acetate) and naturally occurring carnosine dipeptide. The study is interesting and provides unique conclusions and can be recommended for publication in the journal of Biology. The critique of the manuscript is provided below:

Critique:

The major findings of the study should be mentioned in the summary/abstract. (such as AIF-1 expression level and localization changes during differentiation and also under inflammatory stimulus).

The experimental designs including major methods should also be mentioned in Summary/Abstract.

It is suggested to provide a graphical context to table 2 in the form of a figure for a clear understanding of the data. The qPCR data should also be represented in graphs for a clear understanding of the data.

The actin dys(re)organization parameter can be graphed (counting normal and disorganized cells) based on control (7 dps, 21 dps) and PMA (7 dps, 21 dps) and PMA + CAR (7 dps, 21 dps) to clearly represent the findings of the study.

The impact of CAR on AIF-1 localization has not been discussed in detail in the discussion.

The study has no major flaw, but the amount of data generated by this study does not correspond to the level of detail given in the text (introduction and discussion). This is not a critique just a point that needs to be kept while organizing a manuscript.

There are English language errors throughout the manuscript and require thorough proofreading to rectify the mistakes.

Author Response

Critique:

  • The major findings of the study should be mentioned in the summary/abstract. (such as AIF-1 expression level and localization changes during differentiation and also under inflammatory stimulus).

Authors’ response:

We thank the reviewer for this comment to improve the manuscript form. We have revised the abstract accordingly.

  • The experimental designs including major methods should also be mentioned in Summary/Abstract.

Authors’ response:

We thank the reviewer for this comment to improve the manuscript form. We have revised the abstract accordingly, including mentioning the main experimental assays related to the cited results.

  • It is suggested to provide a graphical context to table 2 in the form of a figure for a clear understanding of the data. The qPCR data should also be represented in graphs for a clear understanding of the data.

Authors’ response:

We thank the reviewer for her/his comment. For table 2, we have chosen to maintain this option for brevity and compactness, also for a more streamlined editorial format. Nevertheless, we added the histogram representation of qPCR data reported in table 3, thus changing the text from “table 3” to “figure 5”.

  • The actin dys(re)organization parameter can be graphed (counting normal and disorganized cells) based on control (7 dps, 21 dps) and PMA (7 dps, 21 dps) and PMA + CAR (7 dps, 21 dps) to clearly represent the findings of the study.

We thank the reviewer for this comment letting us give further explanation. We performed the counts of normal vs. disorganized cells in the different experimental conditions (in n = 4 biological replicates). As expected, in 7 dps untreated control, the basal situation of undifferentiated cells shows a monolayer made up of tumor like cells with a “constitutively” high degree of morphological and morphometric heterogeneity and disorganization of the cytoskeleton, both at single-cell level as at monolayer level, i.e., 85%±7 (of total cells counted) appear as disorganized. Contrarily, in 21 dps untreated control, the basal situation of differentiated enterocyte-like cells shows a monolayer with a high degree of morphological and morphometric homogeneity and regular organization of the cytoskeleton, both at single-cell level as at monolayer level, i.e., 100%±4 of the counted cells appear normal. When the counts are performed in PMA-treated undifferentiated monolayers at 7 dps, 71%±8 are counted as disorganized, that means that there’s no significant difference with the respective 7 dps untreated control (85%±7); contrarily, when the counts are performed in PMA-treated differentiated monolayers at 21 dps, 60%±7 of the counted cells appear normal and 40% are counted as disorganized, but we can't help but point out that the number of cells in 21 dps PMA-treated monolayers is “ab initio” considerably reduced (35.7%) with respect to the 21 dps untreated control (100%; see Table 2 in Results): this implies that the number of disorganized cells might be somehow influenced by the serious rearrangement of the monolayer which underwent a hugely reduced number of cells. This is sufficient to make the comparison ineffective between PMA-treated cells at 7 dps (which do not change in number with respect to untreated control) and PMA-treated cells at 21 dps; given that the same argument can be used, for PMA+CAR treatments, in 7 dps monolayers we could count 92%±7 disorganized cells (vs. 85%±7 in 7 dps control), while in 21 dps monolayers we could count only 12%±8 disorganized cells (that is 88%±8 normal, vs. 100%±4 normal in 21 dps control).

  • The impact of CAR on AIF-1 localization has not been discussed in detail in the discussion.

Authors’ response:

We thank the reviewer for this important note. We changed the text accordingly, by adding details as follows: “Despite PMA+CAR treatments did not elicit quantitative differences compared to PMA alone on AIF-1 mRNA/protein levels in our experiments, it is worth noting that CAR showed effects on the localization of the AIF-1 related immunoreactive vescicle-like inclusions in 21 dps enterocyte-like cells; in fact, CAR’s presence seemed to preserve the localization pattern seen in the untreated control, avoiding the perinuclear accumulation observed in cells treated with PMA alone. This evidence will need further investigation, based on the intriguing hypothesis that CAR molecule may act as divalent ion chelator [27], thus interfering with intracellular pathways of AIF-1 as Ca2+-binding protein [12]”.

  • The study has no major flaw, but the amount of data generated by this study does not correspond to the level of detail given in the text (introduction and discussion). This is not a critique just a point that needs to be kept while organizing a manuscript.

Authors’ response:

We are very grateful to the reviewer for this consideration, which we understood precisely in its ameliorative sense.

  • There are English language errors throughout the manuscript and require thorough proofreading to rectify the mistakes.

Authors’ response:

We thank the reviewer for this important hint. We will ask editorial assistance for a better English editing.

Round 2

Reviewer 2 Report

Thanks for the author's reply and effort. The current version of the manuscript is well formatted and well written, so I would recommend accepting it in its present form.